# PROX1 Expression in Resected Non-Small Cell Lung Cancer: Immunohistochemical Profile and Clinicopathological Correlates

**DOI:** 10.3390/medsci13030140

**Published:** 2025-08-17

**Authors:** Evangelia Ntikoudi, Thomas Karagkounis, Konstantinos S. Mylonas, Stylianos Kykalos, Dimitrios Schizas, Ioannis N. Vamvakaris, Ekaterini Politi, Michail V. Karamouzis, Stamatios Theocharis

**Affiliations:** 1First Department of Pathology, National and Kapodistrian University of Athens, 11527 Athens, Greece; 2Society of Junior Doctors, 15123 Athens, Greece; 3First Department of Surgery, General Hospital of Athens “Laiko”, National and Kapodistrian University of Athens, AgiouThoma 17, 11527 Athens, Greece; 4Second Department of Propaedeutic Surgery, General Hospital of Athens “Laiko”, National and Kapodistrian University of Athens, AgiouThoma 17, 11527 Athens, Greece; 5Department of Pathology, Hygeia Hospital, 15123 Athens, Greece; 6Department of Cytopathology, Aretaieion Hospital, Medical School, National and Kapodistrian University of Athens, 11528 Athens, Greece; 7University Pathology Clinic, General and Oncology Hospital “AgioiAnargyroi”, National and Kapodistrian University of Athens, 14564 Athens, Greece

**Keywords:** PROX1, non-small cell lung cancer, adenocarcinoma, squamous cell carcinoma, immunohistochemistry, lymph node metastasis, tumor differentiation, overall survival

## Abstract

**Background/Objectives:** PROX1 (prospero homeobox 1) is a transcription factor involved in lymphangiogenesis and cellular differentiation. Its role in cancer biology appears to be highly context-dependent, with it exhibiting both tumor-promoting and -suppressive functions across various malignancies. Nonetheless, the clinical significance of PROX1 expression in non-small cell lung cancer (NSCLC) remains poorly elucidated. The objective of this study is to evaluate the immunohistochemical expression of PROX1 in NSCLC, specifically in the adenocarcinoma and squamous cell carcinoma subtypes, and to assess its correlation with clinicopathologic features and overall survival (OS). **Methods:** This retrospective study included surgically resected specimens from 121 patients with histologically confirmed NSCLC. PROX1 expression was assessed via immunohistochemistry on formalin-fixed, paraffin-embedded specimens. Staining intensity (graded 0– National and Kapodistrian University of Athens 3) and the percentage of positive tumor cells were recorded. Correlations with histological subtype, tumor characteristics, and OS were analyzed using chi-square tests, one-way ANOVA, and Kaplan–Meier survival analysis with log-rank testing. **Results:** Low PROX1 intensity (level 1) was significantly associated with P63 positivity (*p* = 0.028), while high PROX1 intensity (level 3) correlated with nodal metastasis to station 3 (S3+) (*p* = 0.025). Additionally, alveolar-pattern adenocarcinomas exhibited intermediate PROX1 expression (26–50%) (*p* = 0.010). Although PROX1 positivity did not differ among mucinous and non-mucinous adenocarcinomas (*p* = 0.152), its distribution across defined expression subgroups was statistically significant (*p* = 0.002). Tumors with low PROX1 expression (0–24%) were associated with a larger maximum tumor diameter (*p* = 0.026). PROX1 expression was not independently associated with OS (*p* > 0.05). Factors significantly associated with improved survival included an age < 50 years, female sex, the absence of necrosis, fewer than 10 positive lymph nodes, a lymph node ratio < 0.5, and the absence of extensive nodal involvement in stations 5, 10, 11, and 12. **Conclusions:** Although PROX1 expression is variably associated with specific histologic subtypes and nodal metastases in NSCLC, it does not independently predict overall survival. Its expression patterns suggest a potential role in tumor differentiation and lymphatic spread. Further mechanistic and immunologic studies are warranted to elucidate the functional significance of PROX1 in lung cancer biology.

## 1. Introduction

PROX1 (prospero homeobox 1) is a transcription factor involved in embryonic development, lymphangiogenesis, and cellular differentiation [1,2]. PROX1 is thought to suppress the expression of vascular markers such as laminin, endoglin, intercellular adhesion molecule-1 (ICAM-1), and vascular cell adhesion molecule 1 (VCAM-1), while inducing structural remodeling of blood vessels. Additionally, PROX1 promotes lymphangiogenesis by upregulating key mediators such as lymphatic vessel endothelial hyaluronan receptor-1 (LYVE-1) and fibroblast growth factor receptor 3 (FGFR3) [1,2,3]. In recent years, PROX1 has garnered increasing attention as a key regulatory molecule in cancer biology, owing to its remarkably context-dependent roles across diverse tumor types. In colorectal and gastric cancers, it is predominantly associated with tumor promotion, epithelial–mesenchymal transition, and cellular dedifferentiation [3,4,5,6]. Conversely, in hepatocellular and pancreatic malignancies, PROX1 expression has been linked to more favorable differentiation states and prolonged patient survival [7,8,9,10]. In other tumor types, including breast cancer and oral squamous cell carcinoma, existing data remain conflicting, reflecting its complex and possibly tissue-specific regulatory functions. Notably, PROX1 remains largely unexplored in soft tissue sarcomas and non-small cell lung cancer, representing a significant gap in the current literature [11,12,13].

Ιn a previous comprehensive review, we highlighted the multifaceted and often contradictory roles of PROX1 in neoplasia [14], emphasizing its regulation of pathways like Wnt/β-catenin, Notch, and VEGF-C/VEGFR-3 [15], which are critical to lymphangiogenesis, tumor progression, and immune evasion [16,17,18]. These findings underscore PROX1’s emerging relevance not only as a developmental regulator, but also as a potential diagnostic, prognostic, and therapeutic biomarker across multiple tumor types [19].

The expression profile and clinical relevance of PROX1 in non-small cell lung cancer (NSCLC) remain poorly defined. The current literature offers limited insight into its potential as a predictive tool for patient outcomes. In the present study, we sought to systematically investigate its expression and clinical implications in patients with operable NSCLC.

## 2. Materials and Methods

### 2.1. Study Design and Patient Selection Criteria

This retrospective, observational study aimed to characterize the immunohistochemical expression profile of the homeobox transcription factor PROX1 in resected specimens from patients diagnosed with non-small cell lung carcinoma (NSCLC) between January 2010 and March 2012. The analysis specifically included individuals with histologically confirmed pulmonary adenocarcinoma and squamous cell carcinoma, which represent the two most prevalent histopathological subtypes of NSCLC. Cases diagnosed with small-cell lung carcinoma (SCLC), neuroendocrine tumors, or other rare histologic variants were explicitly excluded to ensure cohort homogeneity and facilitate accurate interpretation of results within the NSCLC group.

Tumor specimens were collected from patients who underwent surgical resection with curative intent at a tertiary Thoracic Diseases Hospital in Athens, Greece. All samples were preserved in formalin-fixed, paraffin-embedded (FFPE) blocks and retrieved from institutional pathology archives. Clinical data, including demographic information, age range, tumor characteristics, stage of disease, and follow-up outcomes, were obtained from electronic medical records and anonymized prior to analysis. Ethical approval was obtained from the local institutional review board, and all procedures conformed to the ethical standards outlined in the Declaration of Helsinki.

### 2.2. Immunohistochemistry and Histopathological Evaluation

PROX1 protein expression was assessed using standard immunohistochemical (IHC) techniques performed on 4 μm sections of FFPE tissue. After deparaffinizationandrehydration, tissue sections were incubated with a primary monoclonal antibody targeting PROX1. Specifically, the protocol included the use of IHC Select^®^ Streptavidin-HRP, prediluted (50 mL), and IHC Select^®^ Secondary Goat anti-Mouse IgG and anti-Rabbit IgG, biotinylated (50 mL), both obtained from Lab Supplies Scientific (Athens, Greece). For the detection of PROX1 expression, a recombinant anti-PROX1 antibody [EPR19273] was used (100 UL; Abcam, Cambridge, UK). Immunoreactivity was semi-quantitatively evaluated by two experienced pulmonary pathologists blinded to clinical outcomes. Discrepancies were resolved by consensus. PROX1 expression was assessed in two distinct dimensions: (1) staining intensity, scored on a four-point ordinal scale (0 = no staining, 1 = weak, 2 = moderate, 3 = strong); and (2) the percentage of positively stained tumor cells, recorded in increments and categorized into predefined groups (0–24%, 25–50%, 51–75%, and >75%).

PROX1 expression levels were subsequently correlated with key clinicopathological features, including histologic subtype, tumor differentiation, lymph node status, presence of necrosis, and other relevant parameters. Specific attention was paid to the anatomical distribution of nodal metastases, particularly with respect to stations defined by the International Association for the Study of Lung Cancer (IASLC) lymph node map [20], according to the 8th Edition Lung Cancer TNM classification [21]. The lymph node ratio (LNR) was also calculated for each patient as the proportion of positive nodes to total nodes examined.

### 2.3. Statistical Analysis

All statistical analyses were conducted using STATA IC15 (StataCorp LLC, College Station, TX, USA). The distribution of continuous variables was first assessed for normality using the Shapiro–Wilk test. Data conforming to a normal distribution were presented as means with standard deviations (mean ± SD), while non-parametric data were expressed as medians with interquartile ranges (Q1–Q3).

Categorical variables were described using absolute frequencies and percentages. Associations between categorical variables were evaluated using the chi-square (χ^2^) test. Comparisons of continuous variables across more than two groups were conducted using one-way analysis of variance (ANOVA), with Bonferroni correction applied for multiple post hoc comparisons to reduce the risk of type I error.

Overall survival (OS), defined as the time from the date of surgery to the date of death or last known follow-up, was analyzed using the Kaplan–Meier method. Survival curves were compared using the log-rank (Mantel–Cox) test, and statistical significance was defined as a two-tailed *p*-value < 0.05.

## 3. Results

### 3.1. Patient Demographics and Tumor Characteristics

The study included a cohort of 145 patients with surgically resected NSCLC. Overall, 24 samples were excluded due to insufficient tumor content after sectioning, degraded tissue quality, or technical IHC failures. The remaining 121 participants were eligible for evaluation. The patients had a mean age of 64.7 years, with a range from 40 to 84 years. The final cohort was predominantly male, comprising 81% males (*n* = 98) and 19% females (*n* = 23).

The cohort consisted of 62 adenocarcinoma and 59 squamous cell carcinoma cases (51.2% and 48.8% respectively, *p* = 0.791). Adenocarcinomas were subclassified histologically into mucinous and non-mucinous variants; 53.2% of adenocarcinomas (33 out of 62 samples) were found to be non-mucinous. Tumors were classified as well-differentiated (8.7%), moderately differentiated (43.5%), and poorly differentiated (47.8%), according to their histologic differentiation. The tumor stage was assessed using the 8th TNM staging edition [21], with 24% of the group categorized as stage I, 36.4% as stage II, and 39.6% as stage III. There were no patients with stage IV disease, as no surgery is recommended at this advanced stage [22]. The average tumor size was 3.5 cm, ranging from 1.2 cm to 8.4 cm.

Surgical procedures included lobectomy (*n* = 78), wedge resection (*n* = 12), and pneumonectomy (*n* = 31). Nodal involvement was evaluated after routinely performed lymph node dissection during surgery. The extent of positive lymph nodes was also recorded; the presence of metastatic cells in the lymph nodes was considered positive involvement. Additionally, the regional lymph node classification for lung cancer staging was used. Special focus was given to nodal stations 3, 5, 10, 11, and 12. The median number of positive lymph nodes per patient was 3, ranging from 0 to 15. The cohort characteristics are summarized in Table 1.

### 3.2. PROX1 Protein Expression and Its Clinicopathological Correlates in NSCLC

The staining patterns after immunohistochemical analysis of PROX1 protein expression throughout the cohort were highly heterogeneous. This condition reflected not only to the proportion of tumor cell positivity, but also PROX1 intensity. PROX1 positivity was detected in the nuclear part of tumor cells, while it was absent in surrounding stromal and inflammatory cells. The staining intensity across the examined NSCLCs ranged from strong, diffuse nuclear (intensity 3) to the total lack of a detectable signal (intensity 0). Representative images of immunostained tumor specimens are shown in Figure 1. According to semi-quantitative scoring, presented in Table 2, weak staining (intensity 1) was the most common pattern, and this expression level showed statistically significant correlations with particular molecular and histological features.

Across the four PROX1 expression groups, the mean patient age extended from 62.85 to 66.80 years (mean ± SD: 65.54 ± 8.62, 63.43 ± 10.3, 62.85 ± 9.73, and 66.80 ± 6.21; *p* = 0.486), with no significant intergroup difference. There was also comparable similarity in the male–female composition of the groups: males, 80.9–90%; females, 10–28.6%; *p* = 0.355. Moreover, the distribution of histological subtypes did not differ with respect to PROX1 levels (*p* = 0.791). Squamous cell carcinoma represented 48.8% and adenocarcinoma represented 51.2% of the total cases, with proportions per group ranging from 40 to 52.4% and from 47.6 to 60%, respectively. However, within subgroups comprising adenocarcinomas harboring an elevated basal level of PROX1, significant enrichment of tumors with an alveolar pattern was found in the intermediate-expression group (26–50%; *p* = 0.010). Table 3 presents the distribution of PROX1 expression (%) in relation to various clinical variables.

Assessment of TNM parameters revealed no significant relationship with PROX1 expression. PROX1 expression in relation to tumor stage, size, and nodal status is presented in detail in Table 4. While reduced expression tended to occur in advanced T stages (T3–T4), this was not statistically significant (*p* = 0.101). Similarly, nodal status exhibited no association with PROX1 expression (*p* = 0.884), with comparable expression patterns throughout the N0 to N2 categories. As already indicated, all patients were M0, precluding the presence of distant metastases. Tumor stage exhibited a non-significant trend (*p* = 0.097), with decreased expression more likely in early-stage disease. Notably, lesion diameter varied substantially by PROX1 expression (*p* = 0.026). More specifically, tumors with a much greater diameter were found in the group with the lowest PROX1 expression (mean 5.36 ± 3.07 cm), whereas the smallest tumors were found in the 26–50% group (mean 3.52 ± 1.71 cm). These findings are indicative of an inverse relationship between lesion size and PROX1 expression. Lymph node metrics—including the number dissected (*p* = 0.811), positive nodes (*p* = 0.435), and lymph node ratio (*p* = 0.958)—did not differ among groups. However, high PROX1 intensity (score 3) was significantly associated with nodal metastases to station 3 (S3) (mean involvement: 0.44 ± 1.33 nodes; *p* = 0.025) (Figure 2).

Marker analysis across PROX1 intensity levels (0–3) identified a significant correlation between weak PROX1 expression (score 1) and P63 positivity (*p* = 0.028). In contrast, no significant differences in the expression of TTF1, CK7, CK5/6, and CK20 were observed across the PROX1 expression groups. Tumor differentiation was not significantly associated with PROX1 expression (*p* = 0.357), although low-grade tumors were more frequently observed in the lowest expression group (0–25%, 52.5%). Similarly, mitotic rate did not correlate significantly with PROX1 levels (*p* = 0.552). Moreover, 46.8% of the adenocarcinomas were mucus-productive, while the vast majority of these mucus-productive tumors showed additional PROX1 expression. However, these associations did not reach any statistical significance (*p* = 0.152). Upon stratification based on PROX1 percentage level, a significant statistical pattern was observed. Specifically, 87.5% of adenocarcinomas in the moderate-PROX1-expression group (26–50%) exhibited mucus production (14 out of 16), compared to only 25%, 30%, and 33.3% in the low (>0–25%), high (51–75%), and very high (76–100%) groups, respectively (*p* = 0.002). This finding indicates a possible increase in mucus-producing phenotypes specifically among tumors exhibiting intermediate PROX1 expression. On the other hand, tumors with very high PROX1 expression (76–100) were found to be more frequently associated with necrosis (90%) compared to those with lower levels (65% to 80%), although not statistically significantly so (*p* = 0.366). A summary of the aforementioned data is provided in Table 5, presenting PROX1 expression (%) in relation to additional clinical variables.

### 3.3. Prognostic Relevance of PROX1 and Other Clinicopathological Parameters

Kaplan–Meier survival analysis revealed no statistically significant association between either PROX1 staining intensity or percentage expression and OS in the studied cohort (*p* > 0.05; Figure 3 and Figure 4). On the other hand, a number of clinicopathological variables exhibited noteworthy prognostic value, as demonstrated in Table 6. Patients younger than 50 years (*p* = 0.021), of female sex (*p* = 0.006), and those without tumor necrosis (*p* = 0.043) had a significant superior survival outcome. Lymph node involvement served as one of the most important prognostic factors. More specifically, subjects with fewer than 10 positive lymph nodes had a markedly superior median OS (60 months) compared to those with 10 or more positive nodes (17 months; *p* < 0.001). Similarly, a lymph node ratio (LNR) < 0.5 corresponded to a median OS of 60 months, while an LNR ≥ 0.5 was associated with only 14 months (*p* = 0.018). Analysis of the site of nodal metastases revealed the negative prognostic implications of extensive lymphatic spread in survival. Involvement of more than four nodes at particular stations was notably correlated with a significant reduction in OS. In detail, nodal involvement at stations 5 (subaortic), 10 (hilar), 11 (interlobar), and 12 (lobar) was associated with an OS of less than 6 months (*p* = 0.003), of 4 months (*p* = 0.038), of 20 months (*p* = 0.002), and of less than 6 months (*p* = 0.003), respectively.

## 4. Discussion

In the present study, we performed a comprehensive clinicopathologic and survival analysis of PROX1 expression in a cohort of NSCLC patients who underwent surgical resection. Our findings underscore a nuanced, context-dependent pattern of PROX1 expression in NSCLC, which varies according to histological subtype, nodal involvement, and morphological features. PROX1 represents a highly prevalent transcription factor that is essential in embryogenesis, organogenesis, cellular differentiation, and lymphangiogenesis [1,2]. Its role in oncology is complex, functioning either as an oncogene or as a tumor suppressor. Its dichotomous behavior is highly dependent on the microenvironment and the molecular signaling pathways, as well as the tissue context [3,11]. For instance, in liver tumors, PROX1 is associated with cell dedifferentiation and prognosis, indicating clear oncogenic involvement [19]. In more detail, there is much evidence suggesting that high PROX1 expression is associated with better prognosis and prolonged survival in cases of hepatocellular carcinoma. In line with these findings, Lim et al. concluded that PROX1 acts as a hepatocyte-specific safeguard repressor, suppressing dedifferentiation and preventing the development of cholangiocarcinoma [7,10].

Additionally, PROX1 plays a key role in neuroendocrine plasticity in prostate cancer, promoting liver metastasis [23]. In contrast, in breast cancer, its exhibits a tumor-suppressive role via its reduced expression due to methylation [24]. Interestingly, its suppression seems to be associated with the progression of breast cancer [25]. Moreover, the involvement of PROX1 in epithelial–mesenchymal transition (EMT) engages it in the progression of colorectal cancer [26]. Similarly, in gastric cancer, its expression correlates with lymphangiogenesis and tumor advancement, reinforcing its potential as a therapeutic target [6,27]. All of these results support PROX1’s pleiotropic function and potential as a context-sensitive biomarker. Crucially, although PROX1’s dual oncogenic and tumor-suppressive functions have been deeply investigated in a variety of malignancies, its relevance in NSCLC remains underexplored.

Notably, in our cohort of NSCLC specimens, the nuclear localization of PROX1—alongside its absence in adjacent stromal and inflammatory cells—strongly underlines its main function as a transcription factor. These observations contribute to the growing body of literature that investigates the role of PROX1 as a pleiotropic transcriptional regulator [28,29]. To the best of our knowledge, no relevant studies have investigated this in human NSCLC tissues, as existing reports are limited to lung cancer cell lines. In light of these findings, we aimed to investigate the association between PROX1 expression and histologic differentiation, lymphatic spread, and tumor growth in NSCLC, as well as to assess its prognostic relevance.

A key finding in our study was the significant correlation between PROX1 expression intensity and P63 positivity. In more detail, the data revealed a higher prevalence of P63-positive cases across weak (level 1) and moderate (level 2) PROX1 intensities. Conversely, P63-negative cases less frequently expressed PROX1 (at any intensity). Given that P63 is a well-known marker of squamous differentiation and the basal cell phenotype [30], these findings might suggest that P63 positivity may be associated with upregulation of PROX1 expression at lower levels, potentially influencing squamous cell carcinoma differentiation. This hypothesis is consistent with prior observations in head and neck and esophageal squamous carcinomas [31,32,33]. Further mechanistic studies are needed to clarify this relationship. On the other hand, adenocarcinomas, especially those with an alveolar growth pattern, were more likely to exhibit higher levels of PROX1. This subgroup showed notable enrichment of tumors with intermediate levels of PROX1-positive cells (26–50%). These findings support the theory that PROX1 supports the preservation of the epithelial identity and architecture in glandular tumors, as observed in gastrointestinal and hepatic cancers [5,9]. Despite the fact that PROX1 expression did not differentiate between mucinous and non-mucinous adenocarcinomas (*p* = 0.152), its distribution among defined expression subgroups was significant (*p* = 0.002). The highest proportion of mucus-producing adenocarcinomas was seen in the group with intermediate PROX1 expression (26–50%). This finding is noteworthy, since it suggests a potential non-linear or context-dependent link between PROX1 activity and secretory differentiation. This trend would imply that intermediate levels of PROX1 expression are sufficient to effectively activate transcriptional pathways associated with mucus production. On the contrary, lower or even higher expression levels may not be able to maintain this phenotype. 

These results may be important for the molecular subclassification of mucinous adenocarcinomas, and therefore warrant further investigation in larger, prospectively annotated cohorts. A notable observation was the robust association between elevated PROX1 staining intensity (level 3) and nodal metastasis to station 3 (S3+), which refers to prevascular and retrotracheal lymph nodes. These anatomical stations are distinctly different from the more frequently involved peribronchial or hilar regions. Metastatic infiltration of prevascular and retrotracheal nodes is frequently indicative of advanced, centrally located disease.

Considering the known role of PROX1 in the process of lymphangiogenesis, particularly via the VEGF-C/VEGFR-3 signaling cascade [2,16,34], its enhanced expression in cases with S3-positivity suggests a potential mechanistic association between PROX1 and selective lymphatic dissemination. This proposition aligns with findings from various other malignancies, in addition to preclinical models, in which PROX1 facilitates the identity of lymphatic endothelial cells and the process of structural remodeling [35]. It is imperative to focus on the anatomical specificity of the nodal burden and its subsequent influence on prognostic outcomes, thereby underscoring the essential importance of precise lymphadenectomy and thorough nodal mapping during the surgical staging process. This hypothesis aligns with findings from other cancer types, as well as preclinical models, where PROX1 facilitates lymphatic endothelial cell identity and remodeling. Emphasis should be given to the anatomical specificity of the nodal burden and its impact on prognosis, emphasizing the importance of meticulous nodal mapping and, during lymphadenectomy, surgical staging. Undoubtedly, nodal dissection of S3 offers more accurate staging, but whether it indeed has a relevant impact on prognosis in cases of NSCLC or not remains controversial [36,37,38]. An inverse relationship was identified between the maximum tumor diameter and PROX1 expression levels in the 0–24% range. Notably, these larger tumors were still surgically resectable, indicating a relatively favorable behavior, despite their advanced size. PROX1 underexpression may be associated with impaired lymphangiogenesis, disrupted apoptotic regulation, and enhanced local invasiveness. This observation is consistent with previous research on colorectal and pancreatic cancers, where silencing or loss of PROX1 expression correlated with more aggressive and high-grade tumors [3,9,39].

Despite the observed association between PROX1 expression and histologic subtypes or lymphatic spread, its presence—whether quantified by intensity or percentage—does not emerge as an independent predictor of OS. Namely, Kaplan–Meier and log-rank analyses failed to demonstrate a statistically significant difference in OS between patients with high versus low PROX1 expression levels. These findings suggest that while PROX1 may influence tumor biology, differentiation, or even tumor progression in cases of NSCLC, it is unlikely to serve as a standalone prognostic biomarker in those patients that have undergone surgical resection with curative intent. It should be highlighted that the prognostic value of PROX1 has been studied in several cancer types, giving conflicting results. In more detail, its overexpression is associated with better prognosis in cases of hepatocellular or pancreatic adenocarcinoma. On the other hand, in cases of colon cancer, esophageal cancer, thyroid cancer, or gliomas, it is linked with unfavorable outcomes [14]. Finally, existing data concerning outcomes in gastric or oral squamous cancer are controversial. According to data extracted from the GEPIA (Gene Expression Profiling Interactive Analysis) database, in cases of NSCLC, PROX1 expression is reduced in tumor tissues compared to normal paired lung tissues. However, tumoral expression levels (low vs. high) do not correlate with OS (*p* = 0.86) [40].

In contrast to PROX1, other common clinicopathological factors represent robust prognostic indicators in NSCLC. Younger age (<50), female sex, the absence of tumor necrosis, and limited lymph node involvement were all linked to improved OS among the subjects included in our cohort. These data align with similar results within the existing literature [41,42]. The favorable prognosis in female patients may reflect hormonal or molecular influences. Tumor necrosis, indicative of hypoxia and poor vascularization, was interestingly associated with worse outcomes. This condition might reflect aggressive tumor behavior (mainly hypoxia-driven) or the development of immunosuppressive tumor environment. Lymph node involvement in stations 5, 10, 11, and 12 had also a negative impact on survival. Patients harboring four or more positive nodes exhibited a median OS under six months. The lymph node ratio (LNR) also emerged as a reliable key predictor, since patients with an LNR < 0.5 had significantly better OS. These data highlight the importance of thorough lymph node evaluation during staging and surgery.

To our knowledge, this represents the largest analysis of PROX1 expression in resectable non-small cell lung cancer (NSCLC). While PROX1 has been studied in other malignancies—with mixed prognostic implications depending on the tumor type—its role in NSCLC remains undefined. In contrast to more extensively characterized biomarkers such as CD44, CD133, and ALDH1 [43,44,45,46], PROX1 has not been systematically evaluated in this context. Given its regulatory role in pathways that are central to tumor progression and immune modulation, we sought to clarify its clinical relevance in NSCLC. Although PROX1 is not currently a therapeutic target, its involvement in VEGF-C/VEGFR-3 signaling [47,48] and lymphatic dissemination suggests its potential role in modifying the tumor microenvironment. We feel that future therapeutic strategies may benefit from targeting PROX1-related pathways, particularly in subgroups with a predisposition for lymphatic invasion.

Several limitations must be acknowledged. First, the study was retrospective in nature and based on a single-institution cohort, potentially introducing selection bias and limiting generalizability. Second, detailed data on neoadjuvant and adjuvant therapies were not uniformly available across the cohort. As a result, we were unable to adjust for potential confounding effects of perioperative treatment on survival outcomes. While this limits our ability to interpret the prognostic value of PROX1 in the context of multimodal therapy, it also reflects the heterogeneity and complexity of real-world clinical practice.

Third, the survival analysis in the present study focused exclusively on overall survival (OS), as recurrence-free survival (RFS) data were incomplete, particularly for patients followed outside of our institution. Additionally, disease-specific survival and other endpoints were not captured, which may have provided further prognostic nuance. Fourth, due to resource constraints and the unfunded nature of the study, additional molecular or immunohistochemical markers—including EGFR, KRAS, ALK, PD-L1, and others—were not analyzed. Similarly, clinical variables such as smoking status were inconsistently recorded and subject to high rates of missingness, precluding meaningful inclusion in multivariable modeling.

Despite these limitations, our study identifies a potentially important prognostic role of PROX1 in resectable non-small cell lung cancer, and provides a foundation for future prospective investigations incorporating molecular profiling, standardized follow-up, and multicenter validation.

## 5. Conclusions

In summary, our study provides novel insights into the differential expression of the transcription factor PROX1 across histological subtypes of NSCLC and its potential involvement in tumor differentiation and regional lymphatic dissemination. The heterogeneity of PROX1 associations across various subgroups underscores its pleiotropic role in cancer biology. While PROX1 expression was significantly associated with markers of squamous and glandular differentiation, as well as with distinct patterns of nodal metastasis, it did not independently correlate with patient OS. These findings suggest that PROX1 may function more as a phenotypic marker of tumor subtype than as a standalone prognostic biomarker. However, future research incorporating molecular profiling, immune landscape characterization, and functional in vivo studies will be essential to elucidate the full role of PROX1in NSCLC pathogenesis and assess its value in clinical stratification or targeted therapy.

## Figures and Tables

**Figure 1 medsci-13-00140-f001:**
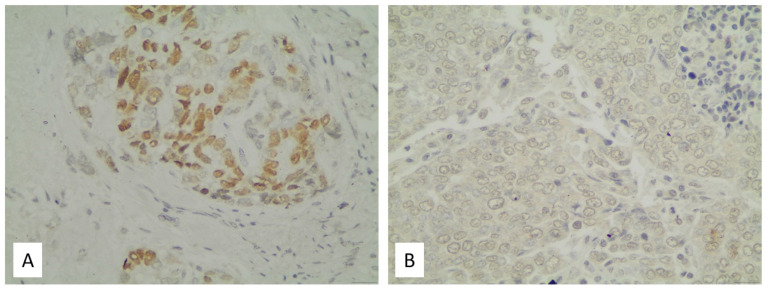
PROX1 expression detected in NSCLC tissues (representative). (**A**): Intense PROX1 staining in a case of adenocarcinoma; (**B**): a low expression pattern in a case of squamous cell carcinoma.

**Figure 2 medsci-13-00140-f002:**
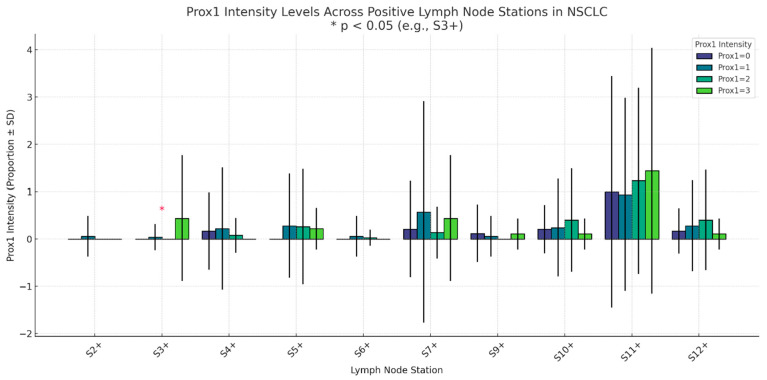
PROX1 intensity levels across LN+ stations.

**Figure 3 medsci-13-00140-f003:**
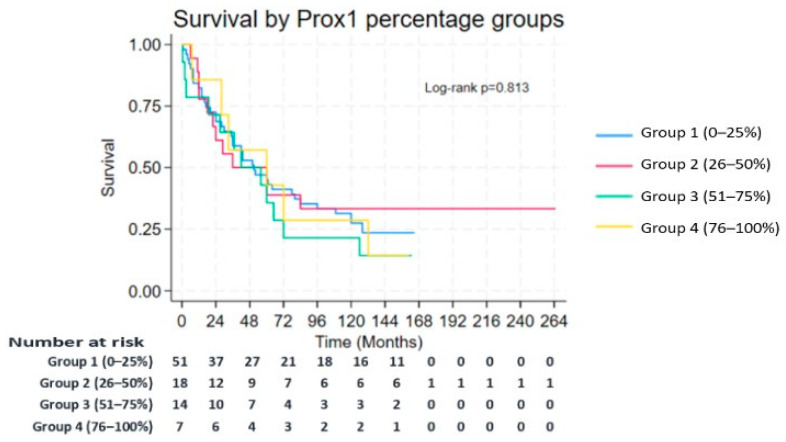
Survival by Prox1 percentage groups.

**Figure 4 medsci-13-00140-f004:**
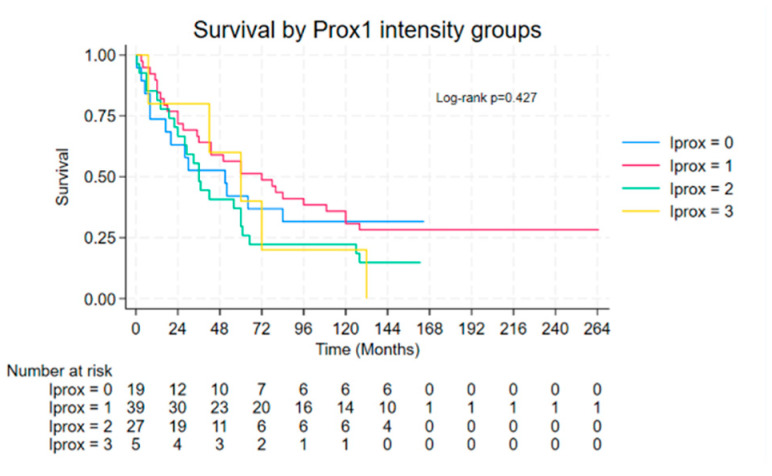
Survival by Prox1 intensity groups.

**Table 1 medsci-13-00140-t001:** Cohort characteristics.

Characteristic	Value
**Total patients (NSCLC, surgically resected)**	145
**Patients eligible after staining**	121
**Mean age (years)**	64.7 (range: 40–84)
**Sex**	Male: 81% (*n* = 98); Female: 19% (*n* = 23)
**Histology**	Adenocarcinoma: 51.2%; Squamous cell carcinoma: 48.8%
**Adenocarcinoma subtype**	Non-mucinous: 53.2% (33/62)
**Tumor grade**	Well-differentiated: 8.7%; Moderately differentiated: 43.5%; Poorly differentiated: 47.8%
**Tumor stage (TNM)**	Stage I: 24%; Stage II: 36.4%; Stage III: 39.6%; Stage IV: 0%
**Mean tumor size (cm)**	3.5 (range: 1.2–8.4)
**Positive lymph nodes**	Median: 3 (range: 0–15)

**Table 2 medsci-13-00140-t002:** PROX1 expression (percentage and intensity).

PROX1 Expression				
Percentage	0–25%	26–50%	51–75%	76–100%
***n*** **(%)**	63 (52.1%)	28 (23.1%)	20 (16.5%)	10 (8.3%)
**Intensity**	**0**	**1**	**2**	**3**
***n*** **(%)**	25 (20.7%)	50 (41.3%)	36 (29.8%)	10 (8.3%)

**Table 3 medsci-13-00140-t003:** PROX1 expression (percentage) and clinical variables.

PROX1 Expression						
Variable	0–25%	26–50%	51–75%	76–100%	Total	*p*-Value
**Age (mean ± SD)**	65.54 (8.62)	63.43 (10.3)	62.85 (9.73)	66.80 (6.21)	64.71 (9.05)	0.486
**Sex**						0.355
- **Male**	51 (80.9%)	20 (71.4%)	18 (90%)	9 (90%)	98 (81.0%)	
- **Female**	12 (19.1%)	8 (28.6%)	2 (10%)	1 (10%)	23 (19.0%)	
**Histology**						0.791
**Squamous cell carcinoma**	33 (52.4%)	12 (42.9%)	10 (50%)	4 (40%)	59 (48.8%)	
**Adenocarcinoma**	30 (47.6%)	16 (57.1%)	10 (50%)	6 (60%)	62 (51.2%)	
**Subtype**						0.607
- **Solid**	52 (83.9%)	16 (57.1%)	13 (72.2%)	8 (80%)	89 (75.4%)	0.054
- **Papillary**	1 (1.6%)	2 (7.1%)	1 (5.6%)	1 (10%)	5 (4.2%)	0.470
- **Micropapillary**	1 (1.6%)	0 (0.0%)	0 (0.0%)	0 (0.0%)	1 (0.8%)	0.823
- **Lepidic**	2 (3.2%)	0 (0.0%)	1 (5.6%)	0 (0.0%)	3 (2.5%)	0.622
- **Alveolar**	4 (6.5%)	9 (32.1%)	2 (11.1%)	1 (10%)	16 (13.6%)	**0.010**
- **Palisading**	1 (1.6%)	0 (0.0%)	0 (0.0%)	0 (0.0%)	1 (0.8%)	0.823
- **Diffuse**	1 (1.6%)	0 (0.0%)	0 (0.0%)	0 (0.0%)	1 (0.6%)	0.823

**Table 4 medsci-13-00140-t004:** PROX1 expression and tumor stage/nodal status.

PROX1 Expression						
Variable	0–25%	26–50%	51–75%	76–100%	Total	*p*-Value
TNM						
**T1a**	1 (1.6%)	0 (0%)	1 (5%)	0 (0%)	2 (1.6%)	0.101
**T1b**	4 (6.3%)	5 (17.8%)	3 (15%)	1 (10%)	13 (10.7%)	
**T1c**	9 (14.3%)	3 (10.7%)	2 (10%)	1 (10%)	15 (12.4%)	
**T2a**	10 (15.9%)	10 (35.7%)	1 (5%)	5 (50%)	26 (21.5%)	
**T2b**	9 (14.3%)	4 (14.3%)	6 (30%)	1 (10%)	20 (16.5%)	
**T3**	13 (20.6%)	5 (17.9%)	5 (25%)	1 (10%)	24 (19.8%)	
**T4**	17 (27%)	1 (3.6%)	2 (10%)	1 (10%)	21 (17.3%)	
**N0**	30 (47.6%)	17 (60.7%)	9 (45%)	5 (50%)	61 (50.4%)	0.884
**N1**	18 (28.6%)	5 (17.9%)	7 (35%)	3 (30%)	33 (27.3%)	
**N2**	12 (19%)	5 (17.9%)	4 (20%)	1 (10%)	22 (18.2%)	
**N3**	0 (0%)	0 (0%)	0 (0%)	0 (0%)	0 (0%)	
**Nx**	3 (4.8%)	1 (3.6%)	0 (0%)	1 (10%)	5 (4.1%)	
**M0**	63 (100%)	28 (100%)	20 (100%)	10 (100%)	121 (100%)	NA
**M1**	0 (0%)	0 (0%)	0 (0%)	0 (0%)	0 (0%)	
**STAGE**						
**IA1**	0 (0%)	0 (0%)	1 (5%)	0 (0%)	1 (0.8%)	0.097
**IA2**	1 (1.6%)	5 (17.9%)	3 (15%)	1 (10%)	10 (8.3%)	
**IA3**	3 (4.8%)	1 (3.6%)	1 (5%)	0 (0%)	5 (4.1%)	
**IB**	4 (6.3%)	6 (21.4%)	0 (0%)	3 (30%)	13 (10.7%)	
**IIA**	7 (11.1%)	3 (10.7%)	10 (50%)	1 (10%)	12 (9.9%)	
**IIB**	17 (27.0%)	6 (21.4%)	7 (35%)	2 (20%)	32 (26.4%)	
**IIIA**	27 (42.9%)	5 (17.9%)	5 (25%)	3 (30%)	40 (33.0%)	
**IIIB**	4 (6.3%)	2 (7.1%)	2 (10%)	0 (0%)	8 (6.6%)	
**Dmax**	5.36 (3.07)	3.52 (1.71)	4.48 (2.65)	4.13(2.52)	4.69 (2.78)	**0.026**
**Nodes**	18.18 (11.54)	17.26 (10.28)	15.60 (8.51)	16.67 (7.98)	17.41 (10.48)	0.811
**Nodes+**	2.41 (4.28)	4.30 (9.10)	2.05 (2.85)	2.11 (3.26)	2.76 (5.55)	0.435
**LN ratio**	0.14 (0.23)	0.15 (0.25)	0.11 (0.16)	0.12 (0.20)	0.14 (0.22)	0.958

**Table 5 medsci-13-00140-t005:** PROX1 expression (percentage) and additional clinical variables.

PROX1 Expression						
Variable	0–25%	26–50%	51–75%	76–100%	Total	*p*-Value
**Necrosis**						0.366
Yes	51 (80.9%)	22 (78.6%)	13 (65%)	9 (90%)	95 (78.5%)	
No	12 (19.1%)	6 (21.4%)	7 (35%)	1 (10%)	26 (21.5%)	
**Mucus**						**0.002**
Yes	9 (14.5%)	14 (50%)	3 (15%)	3 (30%)	29 (24.1%)	
No	53 (85.5%)	14 (50%)	17 (85%)	7 (70%)	91 (75.9%)	
**Differentiation**						0.357
Low	31 (52.5%)	10 (37%)	7 (36.8%)	7 (70%)	55 (47.8%)	
Low/Moderate	9 (15.2%)	6 (22.2%)	2 (10.5%)	0 (0.0%)	17 (14.8%)	
Moderate	13 (22%)	8 (29.6%)	9 (47.4%)	3 (30%)	33 (28.7%)	
Μoderate/High	5 (8.5%)	1 (3.7%)	1 (5.3%)	0 (0.0%)	7 (6.1%)	
High	1 (1.7%)	2 (7.4%)	0 (0.0%)	0 (0.0%)	3 (2.6%)	
**Mitosis**						0.552
Low	1 (2.1%)	1 (5.6%)	0 (0%)	0 (0%)	2 (2.4%)	
Moderate	9 (19.6%)	3 (16.7%)	4 (36.4%)	0 (0%)	16 (19.5%)	
Effective	36 (78.3%)	14 (77.8%)	7 (63.6%)	7 (100%)	64 (78%)	

**Table 6 medsci-13-00140-t006:** OS and clinicopathological prognostic factors.

Variable	Median Survival (Months)	*p*-Value
**Age**		**0.021**
<50	>167 [Q1:56, Q3:167]	
50–59	60 [Q1:12, Q3:120]	
60–69	50 [Q1:22, Q3:165]	
>70	35 [Q1:8, Q3:61]	
**Sex**		
Male	42 [Q1:17, Q3:109]	**0.006**
Female	>167 [Q1:56, Q3:167]	
**Histology**		
Squamous	60 [Q1:22, Q3:167]	0.393
Adenocarcinoma	42 [Q1:17, Q3:120]	
**Nodes+**		
0–9	60 [Q1:24, Q3:131]	**0.001**
10+	17 [Q1:6, Q3:24]	
**LN Ratio**		
0–0.49	60 [Q1:24, Q3:163]	**0.018**
0.50–1.00	14 [Q1:6, Q3:27]	
Stage		
IA	72 [Q1:29, Q3:72]	0.314
IB	60 [Q1:18, Q3:84]	
II	43 [Q1:24, Q3:128]	
IIIA	36 [Q1:14, Q3:84]	
IIIB	12 [Q1:6, Q3:61]	
**Necrosis**		
Yes	43 [Q1:14, Q3:109]	**0.043**
No	72 [Q1:36, Q3:167]	
**Mucus**		
Yes	33 [Q1:18, Q3:96]	0.571
No	56 [Q1:19, Q3:132]	

## Data Availability

The data presented in this study are available on request from the corresponding author. The data are not publicly available due to privacy and ethical restrictions.

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
