# Peer review of "PROX1 Expression in Resected Non-Small Cell Lung Cancer: Immunohistochemical Profile and Clinicopathological Correlates"

_medsci, 2025, doi:10.3390/medsci13030140_

Round 1

Reviewer 1 Report

Comments and Suggestions for Authors

This retrospective study focuses on IHC expression of PROX1 in tumor samples from resected NSCLC patients. I have the following concerns:

  • PROX1 represents a transcription factor that is not particularly well known, therefore a greater level of detail in the introduction would be desirable in order to better clarify its role and intracellular functions
  • the time interval in which the tumor samples were collected must be specified in “Materials and Methods” section
  • Only 121 out of 145 specimens were evaluable; the authors should clarify what this “failure rate” may be due to (line 137)
  • baseline characteristics could be enriched with further information, in particular the type of surgery (lobectomy, wedge resection, pneumonectomy) and any neoadjuvant or adjuvant treatments
  • Correlation between PROX1 other IHC marker expression (such as P63, TTF1, various CK) was explored; if available, it might be interesting to explore correlation between PROX1 and the most common molecular alterations of interest in adenocarcinoma (such as mutation of EGFR, BRAF, KRAS; traslocation/fusion of ALK, ROS1, NTRK), well known predictive factors (such as PD-L1 expression) or clinical features (such as smoking status of patients)
  • Overall survival by PROX1 level of expression does not represent sufficient information to understand the role of this marker; a survival adjustment may be provided based on subsequent treatments that patients received (or not).
  • Furthermore, rather than overall survival, it could be interesting to have disease recurrence-free survival to understand the clinical role of PROX1 expression.

Author Response

Response to Reviewer 1

We would like to thank the reviewer for the thoughtful comments and constructive feedback. We have addressed each point carefully as follows:

Comment 1

PROX1 represents a transcription factor that is not particularly well known, therefore a greater level of detail in the introduction would be desirable in order to better clarify its role and intracellular functions.

Response:
Thank you for your suggestion. The following additional details have been added to our introduction regarding PROX1 (lines 57-72): “PROX1 (prospero homeobox 1) is a transcription factor involved in embryonic development, lymphangiogenesis, and cellular differentiation [1,2]. PROX1 is thought to suppress the expression of vascular markers such as laminin, endoglin, intercellular adhesion molecule-1 (ICAM-1), and vascular cell adhesion molecule 1 (VCAM-1), while inducing structural remodeling of blood vessels. Additionally, PROX1 promotes lymphangiogenesis by upregulating key mediators such as lymphatic vessel endothelial hyaluronan receptor-1 (LYVE-1) and fibroblast growth factor receptor 3 (FGFR3) [1-3]. In recent years, PROX1 has garnered increasing attention as a key regulatory molecule in cancer biology, owing to its remarkably context-dependent roles across diverse tumor types. In colorectal and gastric cancers, it is predominantly associated with tumor pro-motion, epithelial-mesenchymal transition, and cellular dedifferentiation [3-6]. Conversely, in hepatocellular and pancreatic malignancies, PROX1 expression has been linked to more favorable differentiation states and prolonged patient survival [7-10]. In other tumor types, including breast cancer and oral squamous cell carcinoma, existing data remain conflicting, reflecting its complex and possibly tissue-specific regulatory functions. Notably, PROX1 remains largely unexplored in soft tissue sarcomas and non-small cell lung cancer, representing a significant gap in the current literature [11-13].”

Comment 2

The time interval in which the tumor samples were collected must be specified in “Materials and Methods” section.

Response:
We appreciate this suggestion. We have now added the time frame during which the tumor specimens were collected.
(Please see revised lines 89–90: “between January 2010 and March 2012.”)

Comment 3

Only 121 out of 145 specimens were evaluable; the authors should clarify what this “failure rate” may be due to (line 137).

Response:
We agree this requires clarification. Please refer to lines 144-147:“The study included a cohort of 145 patients with surgically resected NSCLC. Overall, 24 samples were excluded due to insufficient tumor content after sectioning, degraded tissue quality, or technical IHC failures. The remaining 121 participants were eligible for evaluation..”)

Comment 4

Baseline characteristics could be enriched with further information, in particular the type of surgery (lobectomy, wedge resection, pneumonectomy) and any neoadjuvant or adjuvant treatments.

Response:
Thank you for your insightful remarks. As described in lines 160-161: “Surgical procedures included lobectomy (n = 78), wedge resection (n = 12), and pneumonectomy (n = 31).”

We also sincerely appreciate the reviewers’ thoughtful suggestion to include information on neoadjuvant and adjuvant therapies. As acknowledged in our revised limitations section (lines 410-415): “Unfortunately, detailed treatment data regarding neoadjuvant and adjuvant therapies were not uniformly available in our retrospective cohort due to institutional limitations in archival documentation during the study period. As a result, we were unable to adjust for potential confounding effects of perioperative treatment on survival outcomes. While this limits our ability to interpret the prognostic value of PROX1 in the context of multimodal therapy, it also reflects the heterogeneity and complexity of real-world clinical practice.We have included this limitation explicitly in the revised discussion section and clarified the retrospective nature of the dataset in the methods.”

We believe the biological and translational insights from our study remain valid and of interest to the field, even in the absence of complete treatment data, and we are grateful for the opportunity to clarify this aspect

Comment 5

Correlation between PROX1 and other IHC marker expression was explored; if available, it might be interesting to explore correlation between PROX1 and the most common molecular alterations of interest in adenocarcinoma (such as EGFR, BRAF, KRAS, ALK, ROS1, NTRK, PD-L1) or clinical features (such as smoking status).

Response:

We thank the reviewer for the insightful suggestion. We fully agree that correlating PROX1 expression with common molecular alterations in lung adenocarcinoma (e.g., EGFR, KRAS, ALK, PD-L1) and clinical variables such as smoking status could provide additional context and biological relevance.

However, the scope of our current study was intentionally focused on characterizing the prognostic role of PROX1 as a single marker in resectable NSCLC. This decision was based both on scientific rationale and practical considerations, as the study was unfunded and resources were not available to perform additional molecular or immunohistochemical analyses. Furthermore, clinical annotation regarding smoking history was inconsistently documented across cases, with a high degree of missingness that precluded reliable inclusion in statistical models.

We have acknowledged these limitations in the revised discussion section (lines 419-424): “Fourth, due to resource constraints and the unfunded nature of the study, additional molecular or immunohistochemical markers—including EGFR, KRAS, ALK, PD-L1, and others—were not analyzed. Similarly, clinical variables such as smoking status were inconsistently recorded and subject to high rates of missingness, precluding meaningful inclusion in multivariable modeling.”

Comment 6

Overall survival by PROX1 level of expression does not represent sufficient information to understand the role of this marker; a survival adjustment may be provided based on subsequent treatments that patients received (or not).

Response:

We thank the reviewer for this valuable suggestion. As described in lines 410-416: “Second, detailed data on neoadjuvant and adjuvant therapies were not uniformly available across the cohort. As a result, we were unable to adjust for potential confounding effects of perioperative treatment on survival outcomes. While this limits our ability to interpret the prognostic value of PROX1 in the context of multimodal therapy, it also reflects the heterogeneity and complexity of real-world clinical practice.

Comment 7

Rather than overall survival, it could be interesting to have disease recurrence-free survival to understand the clinical role of PROX1 expression.

Response:
We agree that recurrence-free survival (RFS) would be a valuable endpoint. However, complete recurrence data (radiologic and/or clinical) were not systematically available for all patients, particularly those who continued follow-up outside our institution. Therefore, we opted to use overall survival as a robust, unbiased endpoint. This limitation is now explicitly stated in the revised Discussion (lines 417-419): “Third, survival analysis in the present study focused exclusively on overall survival (OS), as recurrence-free survival (RFS) data were incomplete, particularly for patients followed outside our institution. Additionally, disease-specific survival and other endpoints were not captured, which may have provided further prognostic nuance.”.        

Reviewer 2 Report

Comments and Suggestions for Authors

In this study, the authors evaluated PROX1 expression in resected non-small cell lung cancer. Several potential points could enhance the current version:

  1. The rationale behind the selection of PROX1 is not sufficiently clear. Numerous well-known lung cancer biomarkers, such as CD44, CD133, and ALDH1, have been extensively studied. The authors should reference relevant publications (e.g., Roudi et al., 2014, 2015, 2016) and provide a rationale for choosing PROX1 over these markers.

  2. The therapeutic relevance of this marker should be clearly defined and discussed.

Author Response

Response to Reviewer 2

We would like to thank the reviewer for their thoughtful comments and constructive feedback. We have addressed each point carefully as follows:

Comment 1

Reviewer:
The rationale behind the selection of PROX1 is not sufficiently clear. Numerous well-known lung cancer biomarkers, such as CD44, CD133, and ALDH1, have been extensively studied. The authors should reference relevant publications (e.g., Roudi et al., 2014, 2015, 2016) and provide a rationale for choosing PROX1 over these markers.

Response:

We would like to than the reviewer for their comment. As described in our revised Discussion (lines 396-403): “To our knowledge, this represents the largest analysis of PROX1 expression in resectable non-small cell lung cancer (NSCLC). While PROX1 has been studied in other malignancies—with mixed prognostic implications depending on tumor type—its role in NSCLC remains undefined. In contrast to more extensively characterized biomarkers such as CD44, CD133, and ALDH1, PROX1 has not been systematically evaluated in this context. Given its regulatory role in pathways central to tumor progression and immune modulation, we sought to clarify its clinical relevance in NSCLC.”

Comment 2

Reviewer:
The therapeutic relevance of this marker should be clearly defined and discussed.

Response:
We agree and have revised the Discussion accordingly. Although PROX1 is not currently a therapeutic target, its involvement in VEGF-C/VEGFR-3 signaling and lymphatic dissemination suggests a potential role in modifying the tumor microenvironment. We propose that future therapeutic strategies may benefit from targeting PROX1-related pathways, particularly in subgroups with a predisposition for lymphatic invasion.

(Please see revised lines402-407in the manuscript: Although PROX1 is not currently a therapeutic target, its involvement in VEGF-C/VEGFR-3 signaling and lymphatic dissemination suggests a potential role in modifying the tumor microenvironment. We feel that future therapeutic strategies may benefit from targeting PROX1-related pathways, particularly in subgroups with a predisposition for lymphatic invasion.

Reviewer 3 Report

Comments and Suggestions for Authors

After reading the manuscript my minor concerns are as follows:

  1. Page 4, Results section: line 137. Please, add information why 24 patients were not eligible for the evaluation in this study? Please, specify the exclusion criteria in a clearer manner.
  2. Page 7, Figure 3. Legend to this figure is an obligation. Please, define the figure 3a, 3b and 3c. What was the main difference between these 3 Figures. Although they illustrated the same data and no statistical significance was observed, why 3 figures were drawn? What was the reason to present 3 figures for the same data? Only one Figure will be enough for illustrating non significant values.
  3. The same holds true for the Figure 4. No legend makes this figure unnecessary, especially, if no significance was reported.
  4. Table 6 on page 13. Please, add the interquartile ranges for median values, as it was mentioned in the Statistics section on page 4. Add information in the legend to this table about the statistical test used when analyze the data. In this Table, several statistically significant comparisons were presented.
  5.  Line 408. Please, remove the patents section

Author Response

Response to Reviewer 3

We thank the reviewer for their careful reading of the manuscript and for these detailed and constructive comments. Below are our point-by-point responses:

Comment 1

Reviewer:
Page 4, Results section: line 137. Please, add information why 24 patients were not eligible for the evaluation in this study? Please, specify the exclusion criteria in a clearer manner.

Response:
We thank the reviewer for his valuable comment. We have now specified that 24 samples were excluded due to insufficient tumor content, degraded tissue quality, or absence of viable material after sectioning.
Please refer to lines 144-147:“The study included a cohort of 145 patients with surgically resected NSCLC. Overall, 24 samples were excluded due to insufficient tumor content after sectioning, degraded tissue quality, or technical IHC failures. The remaining 121 participants were eligible for evaluation..”)

Comment 2

Reviewer:
Page 7, Figure 3. Legend to this figure is an obligation. Please, define the figure 3a, 3b and 3c. What was the main difference between these 3 Figures. Although they illustrated the same data and no statistical significance was observed, why 3 figures were drawn? Only one Figure will be enough for illustrating non-significant values.

Response:
Thank you for your astute observation. Please note that for Figure 3, only one graph has been preserved in the present revised manuscript. Its figure legend reads “Figure 3. Survival by Prox1 percentage groups”

Comment 3

Reviewer:
The same holds true for the Figure 4. No legend makes this figure unnecessary, especially, if no significance was reported.

Response:
Similarly, for Figure 4, only one graph has been preserved in the present revised manuscript. Its figure legend reads “Figure 4. Survival by Prox1 intensity groups”

Comment 4

Reviewer:
Table 6 on page 13. Please, add the interquartile ranges for median values, as it was mentioned in the Statistics section on page 4. Add information in the legend to this table about the statistical test used when analyzing the data. In this Table, several statistically significant comparisons were presented.

Response:
As described in our revised Table 6:“Statistical comparisons were performed using the log-rank (Mantel–Cox) test” Interquartile ranges (Q1-Q3) were also added.

Comment 5

Reviewer:
Line 408. Please, remove the patents section.

Response:
As suggested, the “Patents” section has been removed from the manuscript.